# Nutrient Resorption and C:N:P Stoichiometry Responses of a *Pinus massoniana* Plantation to Various Thinning Intensities in Southern China

Jun Jiang [1,2], Yuanchang Lu [2], Beibei Chen [1,2], Angang Ming [3] and Lifeng Pang [4,*]

[1] College of Forestry, Beijing Forestry University, Beijing 100083, China
[2] Research Center of Forest Management Engineering of State Forestry and Grassland Administration, Beijing 100083, China
[3] Experimental Center of Tropical Forestry, Chinese Academy of Forestry, Pingxiang 532699, China
[4] Institute of Forest Resource Information Techniques, Chinese Academy of Forestry, Beijing 100091, China
[*] Correspondence: plf619@ifrit.ac.cn

**Abstract:** Understanding the responses of C:N:P stoichiometry and nutrient resorption to thinning is essential to evaluate the effects of management practices on biogeochemical cycling in plantation forest ecosystems. However, nutrient resorption and C:N:P stoichiometry do not always respond in the same way to various thinning intensities, and the underlying mechanisms are not well understood. In this study, we aimed to examine the mechanisms underlying the impacts of thinning on C:N:P stoichiometry in a *Pinus massoniana* plantation, focusing on interactions among soils, plant tissues (leaves and litter), and soil properties. We conducted four different thinning treatments to determine the effects of thinning on the C:N:P stoichiometric ratios in leaves, litter, and soil in a *Pinus massoniana* plantation ecosystem. Thinning significantly increased the C, N, and P content of leaves, litter, and soil ($p < 0.05$). The effects of thinning on C:N:P stoichiometry varied strongly with thinning intensity. Specifically, thinning significantly decreased all C:N:P stoichiometry except leaf N:P and litter C:N ($p < 0.05$). The N resorption efficiency (NRE) showed no significant change, but thinning significantly decreased the P resorption efficiency (PRE, $p < 0.05$). This suggests that thinning has inconsistent impacts on N and P cycling in *Pinus massoniana* plantations. In addition, these different responses suggest that soil physicochemical processes play a crucial role in regulating the effects of thinning. Thinning intensity regulates the biogeochemical cycles of C, N, and P in *Pinus massoniana* plantation ecosystems by affecting nutrient resorption and soil physicochemical processes. The inconsistent results obtained can be attributed to the complexities of stand environments and the redistribution of site resources following thinning. Therefore, incorporating the effects of thinning intensity into nutrient cycling models may improve predictions related to achieving long-term forest management strategies.

**Keywords:** thinning intensity; *Pinus massoniana* plantation; C:N:P stoichiometry; nutrient resorption efficiency

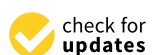



## 1. Introduction

Carbon (C), nitrogen (N), and phosphorus (P), as the most basic constituent elements and the most important nutrients of organisms, play an important role in the balance of elements and the material cycle of ecosystems [1–3]. Plants depend greatly on the balance of multiple nutrients that are essential for plant growth and development [4]. C:N:P stoichiometry can reflect the growth and nutrient status of plants, nutrient limitations, and soil fertility, which play important roles in forest ecosystems. Stoichiometric C, N, and P ratios are widely used as indicators of nutrient limitations on many biotic and abiotic factors [5,6], and their interactions have important implications for the growth characteristics of plants and their adaptation and response to environmental changes [7].

Nutrient reabsorption refers to the process of transferring nutrients to other tissues or directly using them before leaf litter, which reduces the cycle period of nutrients between plants and the environment, which is of great significance for maintaining the nutrient balance and productivity of communities and ecosystems [8]. Plants conserve nutrients and optimize their use for growth and development in forests [9]. The process of plant nutrient resorption effectively improves nutrient utility and makes plant growth less dependent on soil nutrients [10]; it also contributes to nutrient cycling, plant growth, and production. The equilibrium mechanisms maintaining plant C:N:P stoichiometric balance in plant–soil systems are extremely complex [11,12]. Therefore, C:N:P stoichiometry and nutrient resorption are worthy of evaluation at different scales and with multiple different factors, such as vegetation composition [13], nutrient addition [14], and climate change [15]. Such studies may help to enhance our understanding of nutrient cycling and nutrient limitation.

In China, thinning is a common and effective forest management practice that is used to decrease stand density, redistribute resources within a competitive environment, increase resistance to wildfire risk and pests, accelerate the growth of the remaining trees, and improve nutrient circulation processes [16,17]. Previous studies have demonstrated that an appropriate amount of thinning can improve the stand environment, increase tree diameter growth, and maintain and improve soil quality and ecosystem functioning [18–21]. However, these previous studies were limited to projections of stand structure and productivity based on aboveground plant parts; therefore, it remains unclear whether the consideration of how feedback between above- and belowground components influences forest ecosystem functions enhances our ability to predict nutrient stoichiometry and nutrient resorption responses to realistic shifts in thinning conditions [22]. The resulting nutrient inconsistencies are largely mediated by the changes that occur in the stand after thinning, affecting C, N, and P cycling, as they affect the availability of essential nutrients in the forest [23]. Therefore, understanding the response of C:N:P stoichiometry to forest thinning is critical for assessing anthropogenic changes in forest ecosystem function and for developing silviculture measures.

Numerous studies have examined whether thinning modifies soil C, N, and P content and thereby presents the possibility of affecting the temporal patterns of C:N:P stoichiometry within plant–soil systems. For instance, Chen et al. [24] observed that thinning treatments in *Cunninghamia lanceolata* plantations reduced the soil C content by lowering the availability of soil. In contrast, Tian et al. [25] proposed that soil C, N, and P levels increased after thinning in *Pinus massoniana* plantations in China. During the development of a reforested spruce forest, the soil P content did not change significantly under thinning [26]. The responses of C:N:P stoichiometry in different plant tissues to thinning were dramatically different. These studies indicate that C:N:P ratios in plants may differ based on variations in the stand environment, e.g., light, temperature, and moisture conditions [27], understory vegetation [28], and different stand types [29], so the conclusions of previous studies have been inconsistent. Thus, it remains uncertain whether C:N:P stoichiometric links between above- and belowground processes occur in response to thinning.

The nutrient resorption efficiency (NuRE) of plants is greatly limited by the nutrient availability in soils and the nutrient status of the plants. Some studies have demonstrated a positive relationship between NuRE and leaf and soil stoichiometry [30], while other studies have drawn the opposite conclusion [11]. These inconsistent conclusions may be due to the different vegetation types and environmental factors among studies. Therefore, thinning may induce changes in NuRE due to effects on forest community structure, soil properties [31], such as soil micro-organisms or mycorrhizal fungi, and soil and plant nutrient availability and status. However, relatively few publications have explored the inconsistent responses of nutrient resorption and C:N:P stoichiometry to human-induced thinning. Furthermore, few reports have discussed the mechanisms underlying the effects of thinning on NuRE, and the link between soil and plant stoichiometry remains unclear. Therefore, it is necessary to study nutrient cycling under varied thinning intensities for forest management.

Since the 1970s, *Pinus massoniana* forests have been developed on more than 11,300,000 hm$^2$ in southern China due to their high economic and ecological value [32]. Characterized by its excellent natural ability to grow rapidly in diverse soils, including in nutrient-poor conditions, *Pinus massoniana* is one of the most extensively used afforestation trees in southern China [17]. However, *Pinus massoniana* plantations are less productive because high initial planting densities result in a dense canopy and increased competition; these factors subsequently reduce soil fertility and forest productivity. Thinning is the primary widely implemented management practice in such plantation forests.

In this study, we examined the changes in leaf, litter, and soil C, N, and P stoichiometry and nutrient resorption and their relationships in *Pinus massoniana* plantations after 10 years of the application of different thinning intensities in subtropical China. Specifically, the objectives of this research were (a) to examine patterns of leaf, litter, and soil C:N:P stoichiometry in response to different thinning intensities, (b) to identify inconsistent responses to thinning intensity in the C:N:P stoichiometric ratios and NuRE, as well as the thinning-induced changes in soil properties, and (c) to evaluate whether the inconsistent responses are caused by the combined effects of thinning and soil physicochemical and environmental conditions. We hypothesized that (i) the different thinning intensities in the *Pinus massoniana* plantation would cause variation in the C:N:P content of leaves, litter, and soil, which would in turn be reflected in their stoichiometric and NuRE patterns, and that (ii) these inconsistent results could be attributed to the complexities of the stand environment and the redistribution of site resources following thinning, which affect plant stoichiometry and are significantly affected by soil stoichiometry and properties.

## 2. Materials and Methods

### 2.1. Study Area and Thinning Experiment

The study area was located in the Experiment Center of Tropical Forestry and the Guangxi Youyiguan Forest Ecosystem Research Station of the Guangxi Zhuang Autonomous Region (106°39′–106°59′ E, 21°57′–22°19′ N). The region is part of the subtropical monsoon climate zone in China, with rainfall of 1200–1500 mm and an annual mean temperature of 20.5–21.7 °C. The soils at the field site are typical Cambisols according to IUSS Working Group WRB (2006) [33], and the soil layer is over 80 cm deep [17].

The study area was located in a pure *Pinus massoniana* plantation, which was planted in 1993 at a mean initial density of 2500 trees·ha$^{-1}$. These stands also include shrubs such as *Loropetalum chinense*, *Gardenia jasminoides* Ellis, and *Syzygium grijsii*, and the primary herbs include *Dicranopteris linearis* and *Paspalum thunbergii* [17]. Management activities, mainly tending and pruning, were conducted in 1999 and 2003.

In 2007, thinning treatments were applied to four blocks in a randomized block design. Three thinning intensities were determined by the percentage of trees remaining by density following thinning: CK, LT, MT, and HT, with four replications per thinning intensity treatment (Table 1). In each experimental block, four plots (20 × 20 m) corresponding to each treatment were established over an area with a site environment where terrain factors such as elevation, aspect, and slope are approximately the same. The tree species and diameter at breast height (DBH, 1.3 m above the ground) were recorded for all stems ≥5 cm at 2-year intervals from 2008 to 2018. Detailed information on the established plots is presented in Table 1.

### 2.2. Sample Collection and Analysis

In August 2019 (the period of vigorous forest growth), five standard trees were randomly selected in each plot. The standard trees had a diameter and height that were close to the average diameter and height of the stand and were in good growth condition. Five standard trees were selected in each plot, and fresh needles were collected in four different directions in the middle and lower parts of the canopy. All of the litter in the small sample square was harvested by the harvesting method, weighed after mixing, and brought back to the laboratory to be dried in an oven at 80 °C after being mixed.

**Table 1.** Basic information about the four thinning treatments in the *Pinus massoniana* forests. Data are the mean values ± SEs. DBH: diameter at breast height.

| Thinning Treatment | Thinning Intensity (%) | Prethinning Density (tree·ha$^{-1}$) | Postthinning Density (tree·ha$^{-1}$) | Slope (°) | Altitude (m) | Average DBH (cm) | Average Height (m) | Stand Age |
|---|---|---|---|---|---|---|---|---|
| CK | 0 | 1245 | 1231 | 23 | 354 | 14.38 ± 0.36 | 13.55 ± 0.22 | 25 |
| LT | 50 | 1203 | 590 | 25 | 362 | 14.43 ± 1.03 | 12.79 ± 0.31 | 25 |
| MT | 63 | 1116 | 450 | 22 | 273 | 14.55 ± 1.14 | 13.71 ± 0.28 | 25 |
| HT | 80 | 1224 | 225 | 18 | 286 | 14.29 ± 0.34 | 13.11 ± 0.19 | 25 |

Soil samples were collected from five random sampling points to a depth of 20 cm within each of the 20 × 20 m plots at four different treatments (CK, LT, MT, and HT) using a 5 cm drill, following the removal of roots, stones, debris, understory plants, and surface litter. The purity of the litter was determined, and the soil samples were mixed to homogeneity and sieved using a 0.16 mm mesh screen. The samples were then air-dried before the soil chemical analysis. The leaves and soils were sampled in September 2019.

All leaf and soil samples were measured using potassium dichromate-external heating, and after the collected samples were brought back to the laboratory, fresh leaves were rinsed with distilled water and then fixed at 90 °C for 15 min. Fresh leaves and litter samples were dried at 80 °C to a constant weight, ground, and passed through a 0.25 mm sieve before use. The soil organic C (SOC) and the total organic C (OC) of the plant samples were assayed by a modified Walkley–Black acid–dichromate $FeSO_4$ titration method. The total N (TN) was measured using the semimicro Kjeldahl method with a Kjeldahl Autoanalyzer (KDN-102C, Shanghai, China). The total P (TP) in the leaves and soil was measured using the phosphorus vanadium molybdate yellow colorimetric method after $H_2SO_4$–$H_2O_2$ digestion [34]. The water content (SWC) and soil bulk density (BD) were measured using the cutting ring method. The soil available phosphorus (SAP), soil available nitrogen (SAN), and soil pH were determined via NaOH hydrolysis, colorimetry after extraction with $NaHCO_3$, and a standard pH meter in a 1:2.5 soil–water mixture, respectively. The soil organic matter (OM) was determined by the potassium chromate volumetric analysis method [34].

*2.3. Data Calculations and Analysis*

Nutrient resorption efficiency (NuRE) was calculated as the proportion of nutrient resorption during senescence as follows: $NuRE = (1 - \frac{N_{litter}}{N_{leaf}} \times MLCF) \times 100\%$, where NuRE is the efficiency of N or P resorption, $N_{leaf}$ and $N_{litter}$ are the mass of the N or P content in the leaf and litter samples, respectively, and *MLCF* represents the mass loss correction factor (0.745 for conifers trees) [35]. All data are expressed as the mean values ± standard deviations (SDs) of four plots of each treatment. The total C, N, and P content of each soil and leaf sample were used to examine the C:N:P ratios. To test the first hypothesis, a post hoc test for significant differences across different thinning intensities was performed via one-way analysis of variance (ANOVA) followed by Tukey's test at the 5% level. Regression analysis was performed to determine the relationships between leaf, litter, and soil stoichiometry and nutrient resorption. To test the second hypothesis, the Pearson correlation coefficient was used to assess the relationships among soil properties, leaf, litter, and soil N and P content, stoichiometric ratios, and NuRE. The treatments and all stoichiometric combinations were considered additional fixed effects. Redundancy analysis (RDA) was used to determine the degrees of associations between the soil properties and stoichiometry among different thinning intensities. All analyses were performed using R (R Development Core Team).

## 3. Results

### 3.1. Soil Properties and C:N:P Stoichiometry

Soil properties were distinct among the various thinning intensities. The average SWC was significantly higher, following MT and HT, respectively (Table 2, $p < 0.05$). The average pH in the thinning stands was significantly higher, compared with CK (Table 2, $p < 0.05$). Only MT promoted the mean OM, SAN, and SAP, while CK, LT, and HT stands only slightly (insignificantly) (Table 2, $p < 0.05$). By contrast, BD and SOC had no significant difference among the four treatments (Table 2, $p > 0.05$).

**Table 2.** Characteristics of the soil properties in the four thinning treatments in a *Pinus massoniana* plantation. BD: bulk density (g·cm$^{-3}$); SWC: soil water content (%); OM: organic matter (g·kg$^{-1}$); SOC: soil organic matter (g·kg$^{-1}$); SAN: soil available nitrogen (mg·kg$^{-1}$); SAP: soil available phosphorus (mg·kg$^{-1}$). CK: control (0% thinning); LT: light thinning (50% thinning); MT: moderate thinning (63% thinning); HT: heavy thinning (80% thinning). Data are mean $\pm$ SE. Different letters indicate significant differences among different thinning treatments ($p < 0.05$).

| Thinning Treatment | BD | SWC | pH | OM | SOC | SAN | SAP |
|---|---|---|---|---|---|---|---|
| CK | 1.33 $\pm$ 0.10 a | 31.19 $\pm$ b | 4.29 $\pm$ 0.13 a | 10.12 $\pm$ 1.33 b | 14.52 $\pm$ 1.3 a | 120.44 $\pm$ 8.23 b | 0.51 $\pm$ 0.57 b |
| LT | 1.38 $\pm$ 0.09 a | 32.11 $\pm$ b | 4.93 $\pm$ 0.21 b | 12.22 $\pm$ 1.31 ab | 14.88 $\pm$ 1.1 a | 121.15 $\pm$ 13.21 b | 0.54 $\pm$ 0.33 b |
| MT | 1.42 $\pm$ 0.12 a | 35.26 $\pm$ a | 4.51 $\pm$ 0.28 b | 15.40 $\pm$ 1.04 a | 15.12 $\pm$ 1.22 a | 135.28 $\pm$ 11.21 a | 0.79 $\pm$ 0.12 a |
| HT | 1.51 $\pm$ 0.13 a | 36.48 $\pm$ a | 4.7 $\pm$ 0.14 b | 13.01 $\pm$ 1.59 ab | 15.01 $\pm$ 1.01 a | 128.14 $\pm$ 14.23 ab | 0.68 $\pm$ 0.26 ab |

Figure 1 shows the significant variations in the soil C, N, and P content and stoichiometric ratios among the different thinning treatments ($p < 0.05$). The average concentrations of soil C, N, and P in the MT stand were significantly higher than those in the CK, LT, and HT (Figure 1a–c, $p < 0.05$). Soil C content increased by 34.1%, 59.3%, and 47.3%, following LT, MT, and HT, respectively, compared with CK (Figure 1a). LT, MT, and HT increased soil N concentrations 13.0%, 36.2%, and 14.5% more than CK, respectively (Figure 1b). Similarly, LT, MT, and HT increased the soil P concentrations by 56.9%, 99.4%, and 49.4%, respectively (Figure 1c). The average soil C:N, C:P, and N:P ratios in the CK stands were significantly higher than those in the other thinning treatments (Figure 1e,f, $p < 0.05$), and the lowest ratios were observed in the MT stands.

### 3.2. C:N:P Stoichiometry in Leaves and Litter

As shown in Figure 2, thinning had no significant effect on the C content in leaves (Figure 2a, $p > 0.05$), which ranged from 512.46 to 517.83 g·kg$^{-1}$. The average N and P concentrations in the leaves of the CK stand were significantly lower than those of other treatments. Only MT promoted N and P concentrations of the leaves (Figure 2b,c, $p < 0.05$); they ranged from 8.01 to 13.4 g·kg$^{-1}$ and from 0.71 to 0.98 g·kg$^{-1}$, respectively. The leaf C:N and C:P ratios in the CK stands were both significantly higher than those in the other treatments (Figure 2d,e, $p < 0.05$). By contrast, no significant differences in leaf N:P were detected among thinning treatments (Figure 2f, $p > 0.05$).

The C content in the litter exhibited no significant effects under the different thinning treatments and ranged from 400.1 to 450.1 g·kg$^{-1}$ (Figure 3a, $p > 0.05$). Both MT and HT increased litter N and P concentrations (Figure 3b,c, $p < 0.05$). The N and P concentrations of the litter were the highest in the MT stands (Figure 3a,c). The highest litter C:N ratios occurred in the MT stands (Figure 3d); in contrast, HT and MT resulted in a higher litter C:P and N:P ratios (Figure 3e,f, $p < 0.05$).

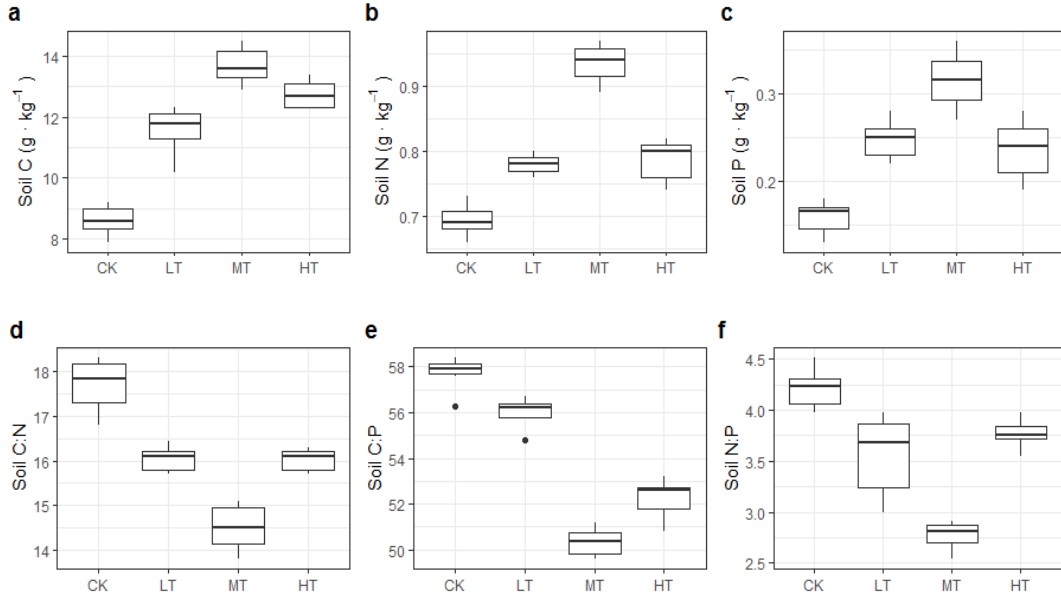

**Figure 1.** Changes in the soil C, N, and P content (**a**–**c**) and soil C:N, C:P, and N:P ratios (**d**–**f**) among four thinning treatments. CK: control (0% thinning); LT: light (50%) thinning; MT: moderate (63%) thinning; HT: heavy (80%) thinning. Same below.

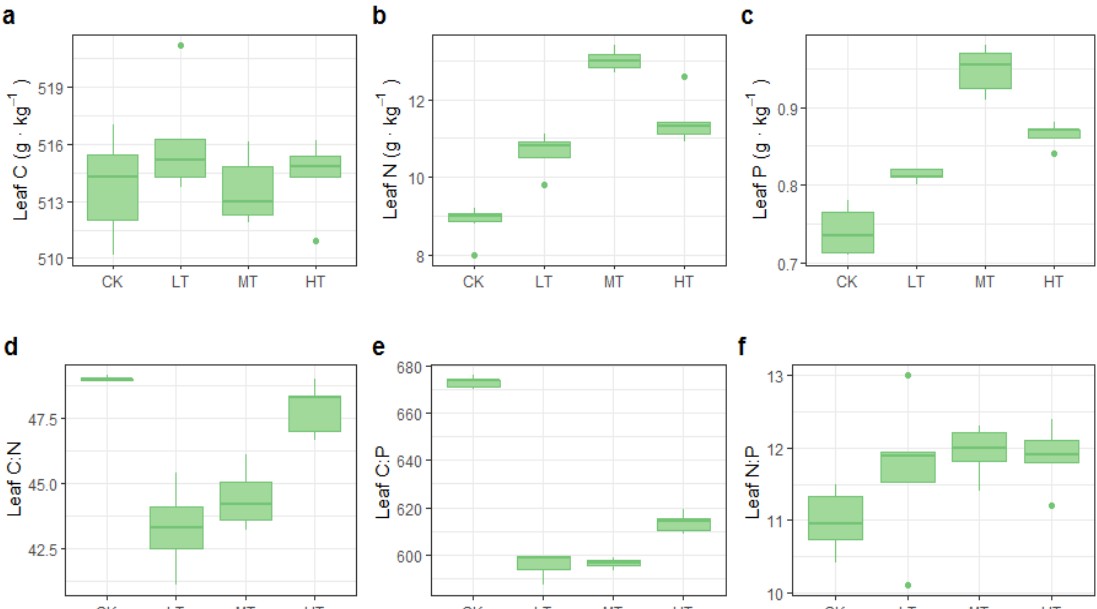

**Figure 2.** Changes in leaf C, N, and P content (**a**–**c**) and leaf C:N, C:P, and N:P ratios (**d**–**f**) among four thinning treatments.

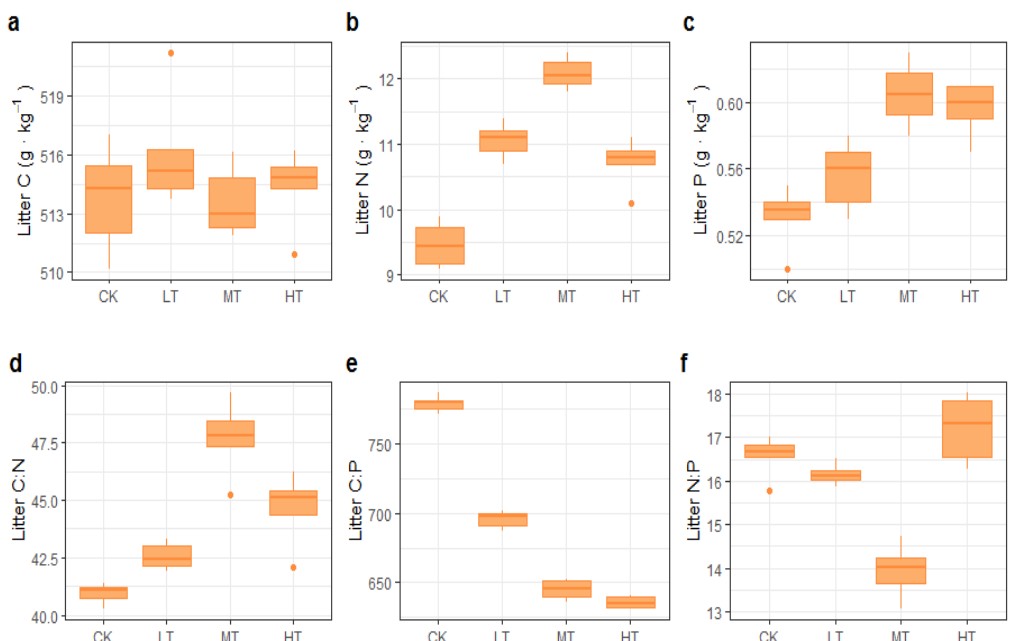

**Figure 3.** Changes in litter C, N, and P content (**a–c**) and litter C:N, C:P, and N:P ratios (**d–f**) among four thinning treatments.

Two-way ANOVAs indicated that thinning significantly affected the leaf N and P, the litter C, N, and P, and the litter C:P and N:P ratios (Table 3). In addition, soil stoichiometry influenced nearly all leaf and litter variables except leaf C, litter C and N, and litter C:N and C:P ratios. Interactions between thinning and soil stoichiometry significantly affected the leaf N and the N:P ratio (Table 3).

**Table 3.** F and P values for the effects of thinning and soil stoichiometry on plant and soil C:N:P stoichiometric characteristics in a *Pinus massoniana* forest plantation. The soil stoichiometry refers to the same element as that in a given row.

| | Thinning | | Soil Stoichiometry | | Thinning × Soil Stoichiometry | |
|---|---|---|---|---|---|---|
| | **F** | ***p*-Value** | **F** | ***p*-Value** | **F** | ***p*-Value** |
| Leaf C | 3.5 | 0.45 | 22.4 | 0.42 | 36.66 | 0.79 |
| Leaf N | 45.2 | <0.05 | 51.3 | <0.01 | 8.64 | <0.001 |
| Leaf P | 36.7 | <0.001 | 44.3 | <0.01 | 15.58 | 0.63 |
| Litter C | 0.54 | <0.05 | 21.6 | 0.34 | 10.32 | 0.44 |
| Litter N | 31.2 | <0.01 | 8.7 | 0.67 | 25.55 | 0.16 |
| Litter P | 26.5 | <0.01 | 31.4 | <0.05 | 48.26 | 0.11 |
| Leaf C:N | 10.2 | 0.73 | 61.3 | <0.001 | 27.46 | <0.001 |
| Leaf C:P | 15.1 | 0.56 | 37.5 | <0.05 | 13.22 | 0.22 |
| Leaf N:P | 12.6 | 0.36 | 49.4 | <0.05 | 2.73 | <0.05 |
| Litter C:N | 26.4 | 0.19 | 12.2 | 0.44 | 14.63 | 0.34 |
| Litter C:P | 41.2 | <0.01 | 36.1 | 0.18 | 9.63 | 0.17 |
| Litter N:P | 25.7 | <0.01 | 55.3 | <0.05 | 28.77 | 0.41 |

In addition, there were significant positive correlations between the soil and leaf C, N, and P content ($p < 0.05$) (Figure 4), and leaf N and P exhibited significant correlations with soil N and P, respectively (Figure 4b,c). Similarly, the leaf C:P and N:P ratios were correlated with the soil C:P and N:P ratios (Figure 4f).

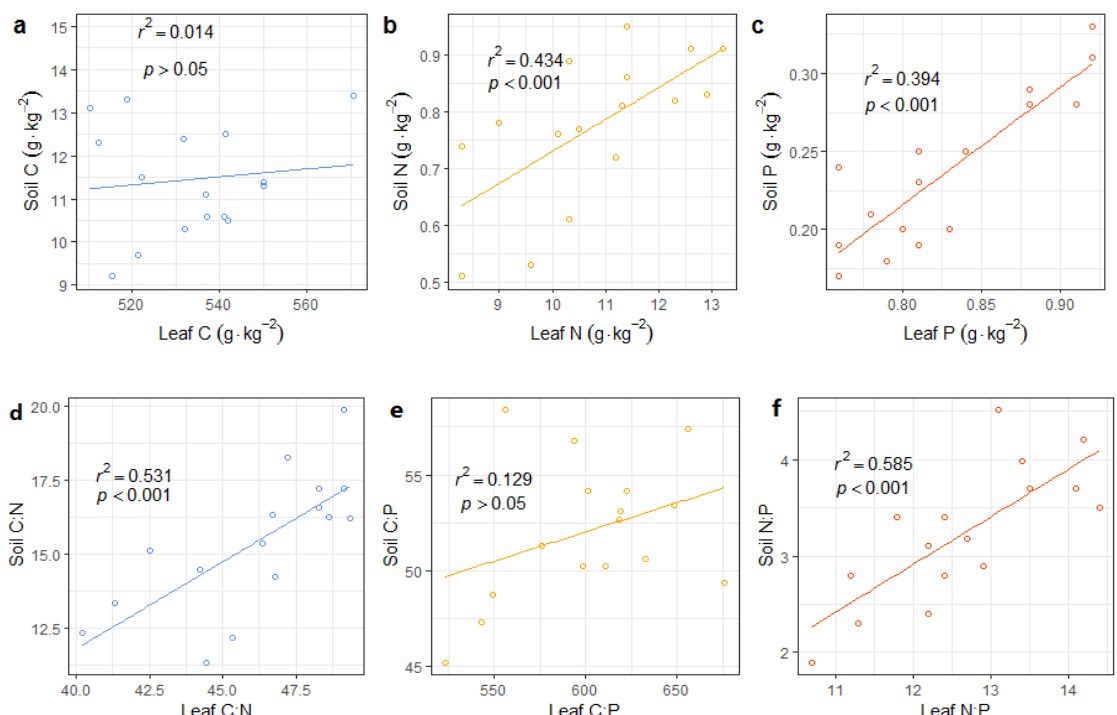

**Figure 4.** Correlations of leaf C, N, and P content(**a–c**) and C:N, C:P, and N:P ratios(**d–f**) with soil C, N, and P content and stoichiometry.

### 3.3. NRE, PRE, and Response to Leaf, Litter, and Soil Stoichiometry

The differences in NRE among various thinning intensities were not significant (Figure 5a, $p > 0.05$), although a slight increase was observed under thinning. The NRE ranged from 34.7% to 35.9%. The average PRE in the CK was significantly higher than others (Figure 5a, $p < 0.05$), with the PRE ranging from 45.2% to 55.1%. MT and HT resulted in a further increase in the NRE:PRE ratio, compared with CK and LT (Figure 5b, $p < 0.05$).

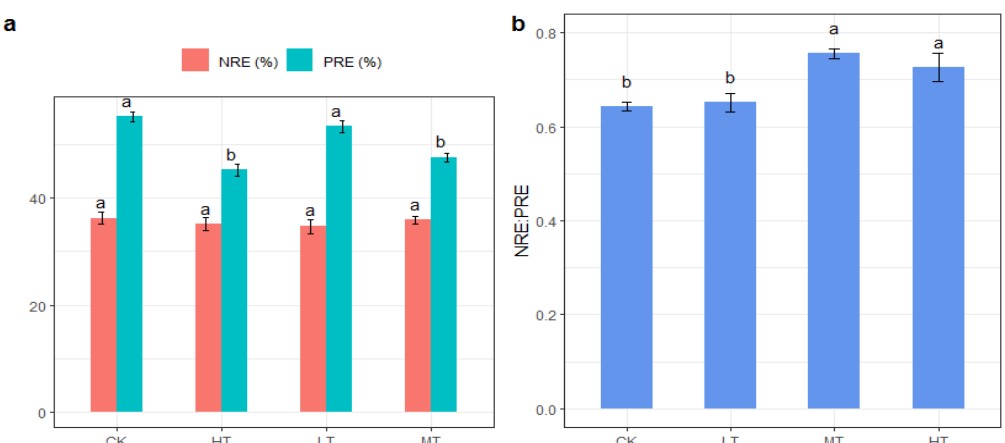

**Figure 5.** Changes in NRE, PRE (**a**), and NRE:PRE (**b**) among different thinning treatments. Different letters indicate significant differences among different thinning treatments ($p < 0.05$).

The NRE showed nonsignificant differences among leaves, litter, and soil, except in litter N and leaf N:P (Figures A1 and A2). The NRE was significantly negatively correlated with litter N (Figure A1, $p < 0.05$) and positively correlated with leaf N:P (Figure A2, $p < 0.05$). The PRE was significantly negatively correlated with the leaf P, litter C and N ($p < 0.05$), soil C, N, and P content ($p < 0.001$), and the litter C:N ratio (Figure A3, $p < 0.05$), and was significantly positively correlated with the leaf C:N, C:P, and N:P ratios ($p < 0.05$),

the litter C:P ratio ($p < 0.05$), the soil C:P ratio ($p < 0.05$), and the soil N:P ratio ($p < 0.001$, Figure A4).

### 3.4. Relationships between Soil Properties, Stoichiometry, and NuRE

The redundancy analysis (RDA) identified the effects of the soil nutrient content, stoichiometric ratios, and soil properties (BD, SWC, pH, SAN, SAP, SOC, and OM) on the dominant orders under different thinning intensities (Figure 6). The RDA showed that Axis 1 accounted for 53.61% of the variation and that Axis 2 accounted for 23.51%. The soil C, N, and P content was significantly positively correlated with SAP, SOC, OM, and pH. Excluding the soil C:N, SWC, and SAP, the soil properties, nutrient content, and stoichiometric ratios were significantly negatively correlated with BD, while the soil C:P and N:P ratios were positively correlated with SWC, OM, SOC, and pH (Figure 6).

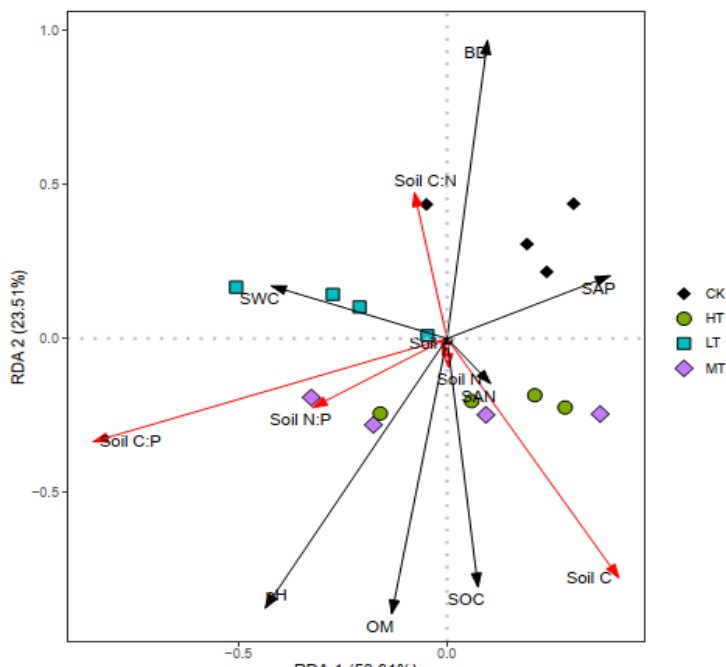

**Figure 6.** Ordination plots of the results of the redundancy analysis (RDA) identifying the relationships between soil properties (black arrows) and soil C:N:P stoichiometry (red arrows) among different thinning treatments.

Pearson correlation analysis was used to further analyze the relationships among the leaf, litter, and soil properties (BD, SWC, SAN, SAP, OM, SOC, and pH) and the stoichiometric ratios and NuRE (Table 4). The C, N, and P content in leaves was significantly negatively correlated with soil BD ($p < 0.05$) and significantly positively correlated with SWC ($p < 0.05$). The leaf and litter N and P were significantly negatively correlated with pH ($p < 0.05$). Leaf N and P and litter N were positively correlated with OM ($p < 0.05$). Leaf and litter C were significantly positively correlated with SOC ($p < 0.05$). Litter N and P had significant positive correlations with SAN and SAP ($p < 0.05$). The litter C:N ratio had a significant positive correlation with SOC ($p < 0.01$). The leaf C:P was significantly positively correlated with SAP ($p < 0.05$). The correlation analysis revealed no correlations between soil properties and the NRE ($p > 0.05$). The PRE had significant positive correlations with soil BD ($p < 0.01$) and was negatively correlated with soil pH ($p < 0.01$) and SAP ($p < 0.05$).

**Table 4.** The correlation coefficients between nutrient content, soil properties, stoichiometric ratios in leaves and litter, and NuRE after thinning. *** $p < 0.001$; ** $p < 0.01$; * $p < 0.05$.

|           | BD          | pH          | SWC         | OM          | SOC         | SAN         | SAP         |
|-----------|-------------|-------------|-------------|-------------|-------------|-------------|-------------|
| Leaf C    | −0.673 **   | −0.534      | 0.532 ***   | 0.376       | 0.425 *     | 0.216       | 0.102       |
| Leaf N    | −0.397 *    | −0.541 *    | 0.613 **    | 0.446 **    | 0.254       | 0.365       | 0.053       |
| Leaf P    | −0.558 **   | −0.818 ***  | 0.684 ***   | 0.546 **    | 0.013       | 0.485       | 0.542       |
| Litter C  | −0.344 *    | −0.256      | 0.634 *     | 0.367       | 0.413 **    | 0.264       | 0.102       |
| Litter N  | −0.611 **   | −0.489 *    | 0.715 **    | 0.706 **    | 0.395       | 0.517 *     | 0.097 *     |
| Litter P  | −0.734 ***  | −0.738 ***  | 0.687 **    | −0.213      | 0.275       | 0.657 *     | 0.324 ***   |
| Leaf C:N  | 0.056       | 0.113       | −0.036      | 0.156       | 0.116       | −0.089      | −0.067      |
| Leaf C:P  | 0.186       | 0.624 **    | −0.162      | 0.084       | 0.204       | −0.495      | −0.617 **   |
| Leaf N:P  | 0.326       | 0.401 **    | −0.203      | 0.526 **    | 0.068       | −0.107      | −0.232      |
| Litter C:N| 0.184       | 0.058       | −0.205      | 0.534 *     | 0.335 *     | −0.016      | −0.048      |
| Litter C:P| 0.626 ***   | 0.724 **    | −0.748 **   | 0.220       | 0.243       | −0.611 **   | −0.107      |
| Litter N:P| 0.703 **    | 0.806 **    | −0.824 ***  | 0.617 **    | 0.513       | −0.724 **   | −0.265      |
| NRE       | 0.212       | 0.156       | 0.035       | 0.104       | 0.097       | 0.048       | 0.205       |
| PRE       | 0.638 **    | −0.679 **   | 0.322       | 0.103       | 0.054       | −0.059      | −0.334 *    |

## 4. Discussion

### 4.1. Effects of Thinning Intensity on Soil C, N, and P Content and Stoichiometry

Thinning significantly increased the soil C, N, and P content, and we detected significantly higher soil C, N, and P content in MT than in the other thinning treatments. We speculate that this difference following thinning can be attributed to improvements in the microclimate and the soil substrates. Thinning is one of the main anthropogenic activities that influence the structure and composition of aboveground vegetation, thereby improving the microclimate (i.e., soil temperature and moisture conditions) and the quantity and quality of litter. Moreover, compared with those in the CK plots, the C, N, and P concentrations in the MT plots were significantly higher. A similar trend was found for the average OM, SWC, SAN, and SAP after thinning. This may have occurred because moderate thinning improved resource availability for the remaining trees, hence promoting litter decomposition and a more favorable microclimate. This result demonstrates that thinning had positive effects on soil quality for a certain period of time.

The responses of soil C:N:P stoichiometry to thinning intensity have important implications for the development of sustainable forest management strategies. The soil C:N:P stoichiometry of the *Pinus massoniana* plantation was weakly regulated by the thinning intensity. Our results show that moderate thinning decreased the C:N, C:P, and N:P ratios, revealing that thinning could promote OM decomposition and N mineralization, as well as P availability in MT stands. Thus, the inputs of litterfall and root exudates may partly compensate for N loss, resulting in a decreased C:N ratio under moderate and heavy thinning. Alternatively, these patterns of variation might be related to improvements in the vegetation and soil environment, i.e., higher SWC, SAN, and SAP; this would be similar to the results of the study reported by Tian et al. [33].

The RDA results indicate that soil C:N:P stoichiometry between different thinning intensities could be well ascribed to changes in soil properties. Soil C, N, and P content was significantly positively correlated with SAP, SOC, and OM, and the soil N and C:P and N:P ratios were closely correlated with pH (Figure 6) because SAP, SOC, and OM can be directly absorbed by plants; therefore, these factors can reflect the dynamics of soil supply very sensitively [27]. Soil BD is a particularly interesting factor in our study. BD was negatively correlated with soil properties, nutrient content, and stoichiometric ratios, except the soil C:N, SWC, and SAP, because a higher BD is not beneficial to soil aeration, water retention, soil nitrogen mineralization, or P availability [36,37]. The soil C:P and N:P ratios were positively correlated with SWC, OM, SOC, and pH, which indicated that soil properties can function as catalysts for soil nutrient availability [38].

### 4.2. Effects of Thinning Intensity on Leaf and Litter C, N, and P Content and Stoichiometry

The N and P content of leaves and litter was strongly regulated by the thinning intensity. By increasing thinning intensity, the effects on the N and P content of leaves and litter became more positive, and MT resulted in the highest N and P concentrations in

leaves and litter; however, the effects of the different treatments on the C content of leaves and litter were not significantly different (Figure 3a,d). The increased leaf N and P content may have arisen from thinning-induced reductions in competition and improvements in the availability of nutrients and space for the retained trees, which would promote stand growth. This is a growth adaptation strategy in which fast-growing plants require higher amounts of N and P to combine their RNA to support rapid growth [39]. Similar to the situation in litter, the increase in litter N and P content with increased thinning intensity may have been because of the positive plant–soil feedback from litter decomposition. Our results show lower C:N and C:P ratios but higher N:P ratios in leaves following thinning. The inconsistent responses of leaf C:N, C:P, and N:P ratios to thinning intensity may have resulted from the significant increases in N and P content and the unchanged C content. These differences in thinning effects may be explained by the fact that C:N and C:P ratios are correlated with plant growth rates [6], indicating that thinning may affect the growth of remaining trees.

Furthermore, the leaf N:P ratio is considered an indicator of the relative nutrient limitation of plants [7]. In theory, a leaf N:P ratio >16 indicates P limitation, and an N:P ratio <14 indicates N limitation [5]. Our results on leaf N:P ratios suggest that *Pinus massoniana* forests are N-limited. In addition, this study revealed that interactions between thinning treatments and soil stoichiometry significantly affected leaf N and the N:P ratio, indicating that soil properties may be associated with the plant nutrient status caused by thinning. We found that the N and P content and stoichiometry in leaves were significantly correlated with soil C, N, and P content and stoichiometry. This finding suggests that soil nutrient supply is an important factor responsible for changes in plant nutrient status, consistent with previous studies. [14,24]. Thus, we speculate that the inconsistent stoichiometric responses in the leaves may be a result of the interaction between thinning and environmental variations.

Litter usually mediates plant–soil feedback because it releases soil nutrients as it decomposes. Therefore, litter may be the link between plant and soil stoichiometry. A previous study found a correlation between the responses of litter C:N:P stoichiometry and thinning practices [40]. We found that the effects of thinning intensity on the litter stoichiometric ratios were stronger than those on the soil stoichiometric ratios (Figure 3e), which suggests that thinning affects these ratios through similar mechanisms; for example, the litter produced by thinning decomposes and releases nutrients quickly [41].

### 4.3. Effects of Thinning Intensity on NRE and PRE

The average NRE and PRE values in *Pinus massoniana* plantations were found to be 62.1% and 64.9% lower than global terrestrial forests, respectively [35]. The NRE was not significantly different among the different thinning treatments, which does not support our first hypothesis. Our results show that the NRE was significantly positively correlated with leaf N:P, indicating that leaf N:P may be the main factor determining nutrient resorption [42]. However, the NRE had a negative relationship with litter N, possibly because thinning stimulated the decomposition of litter N through abiotic pathways. In other words, thinning activities might change litter structures through the microclimate-driven acceleration of decomposition rates. These effects may explain why the correlation between plant and soil stoichiometry varies with thinning intensity from another perspective.

Interestingly, the PRE in the CK stands was significantly higher than that in the thinned stands. In this study, the PRE was significantly negatively correlated with the leaf P, litter C and N ($p < 0.05$), soil C, N, and P content, and the litter C:N ratio, which supports our first hypothesis. This change may be closely associated the significant increase in soil P and leaf P and SAP content under thinning (Figures 3c and 4c, Table 2), which would reduce phosphorus uptake by senescent leaves due to adequate phosphorus supply. Both MT and HT lead to a higher NRE:PRE ratio; this is mostly because more N than P was reabsorbed. This evidence supports the "relative resorption hypothesis", which states that, theoretically, when N or P is limited, plants resorb proportionally more N or P, which allows them to

balance their nutrient status. This evidence of the relative changes in the NRE and PRE suggests that the *Pinus massoniana* plantation in our study exhibits N limitation.

The effects on stoichiometry and NuRE have been found to be inconsistent in forests [22]. For example, the removal of trees through thinning often results in changes in microclimate conditions, as the canopy closure is reduced, so soil uptake of solar radiation is increased [43]. Site-specific microclimatic conditions are known to affect biogeochemical C, N, and P cycles and might also affect the responses of these cycles to thinning. However, there were different degrees of correlation between nutrient content, stoichiometric ratios, and NuRE, and other studies have shown disparate results [44]. These findings add uncertainty to estimates related to potential future changes in forest management practices.

*4.4. Implications for Future Forest Management Practices*

Knowing the effects of thinning intensity on the C:N:P stoichiometry of *Pinus massoniana* plantation ecosystems helps us predict plant–soil feedback. In our study, the variations in nutrient content, stoichiometric ratios, nutrient resorption, and soil properties demonstrate that thinning induces significant increases in nutrient content, promotes nutrient cycling, and improves soil fertility. Among the four thinning treatments, the highest nutrient levels were observed under MT, which indicates that this treatment can be recommended for *Pinus massoniana* plantations. In addition, soil C:N:P stoichiometry exhibited differential responses to thinning. Differences in soil properties and in the soil environment within plant–soil systems may affect the response of stoichiometric ratios to thinning. For example, the soil BD and pH have been suggested to be sensitive measures of changes in soil status and useful indicators of soil processes. Our results show that thinning intensity significantly affects plant and soil C:N:P stoichiometry and its association. This finding supports our second hypothesis and indicates that the effect of management practices on the relationship between C, N, and P content and their stoichiometric ratios is a joint result of multiple factors, including the soil type, plant species, and management practice. In addition, thinning may trigger plant nutrient conservation strategies and may alter plant nutrient use efficiency levels, as well as the degree to which plant growth depends on soil nutrients [45,46]. These thinning-induced changes in C:N:P stoichiometry play a key role in regulating plant–soil feedback and maintaining plant stoichiometric balance. Integrating the effects of thinning on C:N:P stoichiometry into forest management and restoring plantation forests on the basis of field observations may enhance the productivity of Chinese *Pinus massoniana* plantations.

Most studies of C:N:P stoichiometry were conducted in temperate plantations in eastern Asia. More experiments are needed to understand the impact of thinning on subtropical plantations. Moreover, fertilization experiments are needed to provide mechanistic insight into nutrient limitation and to differentiate between N and P limitation following thinning in long-term studies. In addition to climate, vegetation type, site conditions, and management history may also regulate the response of C:N:P stoichiometry to thinning. Replanting N-fixing tree species in pure plantations could balance soil N availability [45,47]. Nevertheless, well-designed N addition experiments might yield insights into the effects of thinning on plant resorption and nutrient cycling, which would help to evaluate the effects of thinning in plantations [48,49].

## 5. Conclusions

Thinning is an important silvicultural measure that influences forest ecosystem C, N, and P content, as well as their stoichiometric relationships. Our findings show that, generally, moderate thinning significantly increased C, N, and P content in green leaves, litter, and soil, except for C content in green leaves. Thinning significantly reduced the C:N and C:P ratios of green leaves, the C:P and N:P ratios of litter, and the C:N, C:P, and N:P ratios of soil, indicating that thinning has an impact on the biogeochemical cycles of C, N, and P in forest ecosystems. Thinning was closely associated the PRE, which provides further evidence of N limitation in *Pinus massoniana* plantations in our study region. The

differential responses of soil properties to thinning and their function as a link between plants and soil suggest that soil physicochemical processes play a crucial role in regulating thinning effects. Forest management practices that are focused on combining aboveground and belowground processes in plantation forests could help us to better understand the inconsistent responses to thinning. Considering the effects of thinning intensity on nutrient cycling, such as alleviating nutrient limitations, replanting certain tree species, restoring degraded stands, or maximizing forest productivity, may improve predictions of how short- and long-term forest management goals can be achieved in *Pinus massoniana* plantations in southern China.

**Author Contributions:** All the authors conceived the study, collected the data, performed statistical analysis, and helped to draft the manuscript. All authors have read and agreed to the published version of the manuscript.

**Funding:** This research was funded by the National Natural Science Foundation of China (No. 31901306) and the National Natural Science Foundation of Guangxi Province (2020GXNSFAA297208).

**Institutional Review Board Statement:** Not applicable.

**Informed Consent Statement:** Not applicable.

**Data Availability Statement:** Not applicable.

**Acknowledgments:** We thank Xianqiang Zhang and all individuals from Experimental Center of Tropical Forestry for their help with fieldwork.

**Conflicts of Interest:** The authors declare that they have no competing interests.

## Appendix A

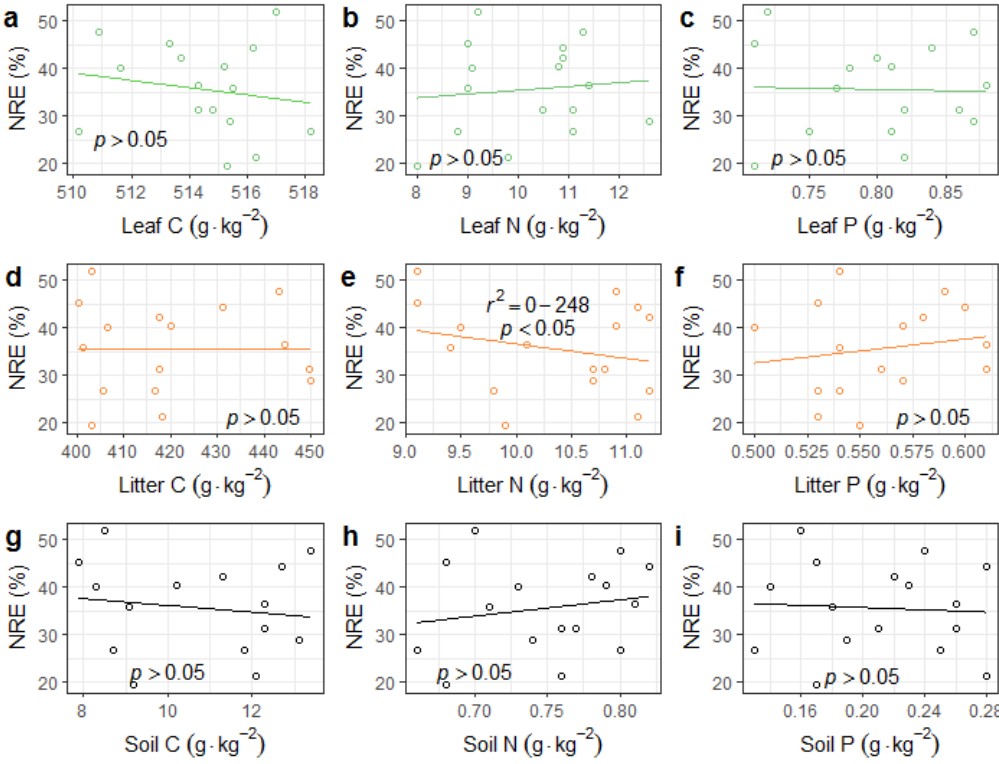

**Figure A1.** Relationships between NRE (N resorption efficiency) and leaf (**a**–**c**), litter (**d**–**f**), and soil (**g**–**i**) C, N, and P contents.

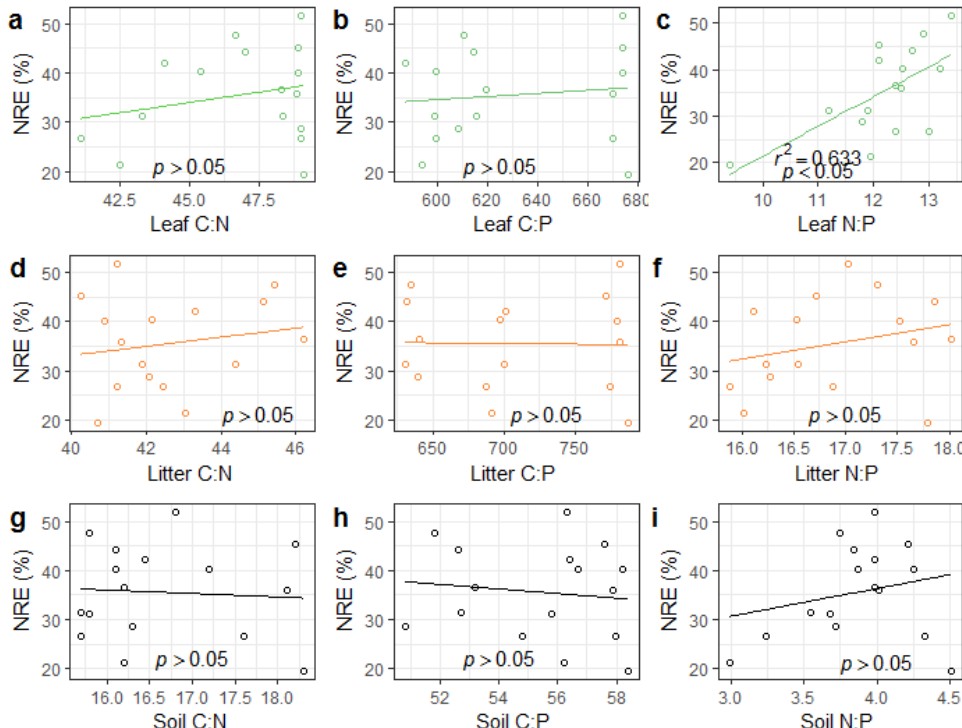

**Figure A2.** Relationships between NRE (N resorption efficiency) and leaf (**a**–**c**), litter (**d**–**f**), and soil (**g**–**i**) stoichiometry.

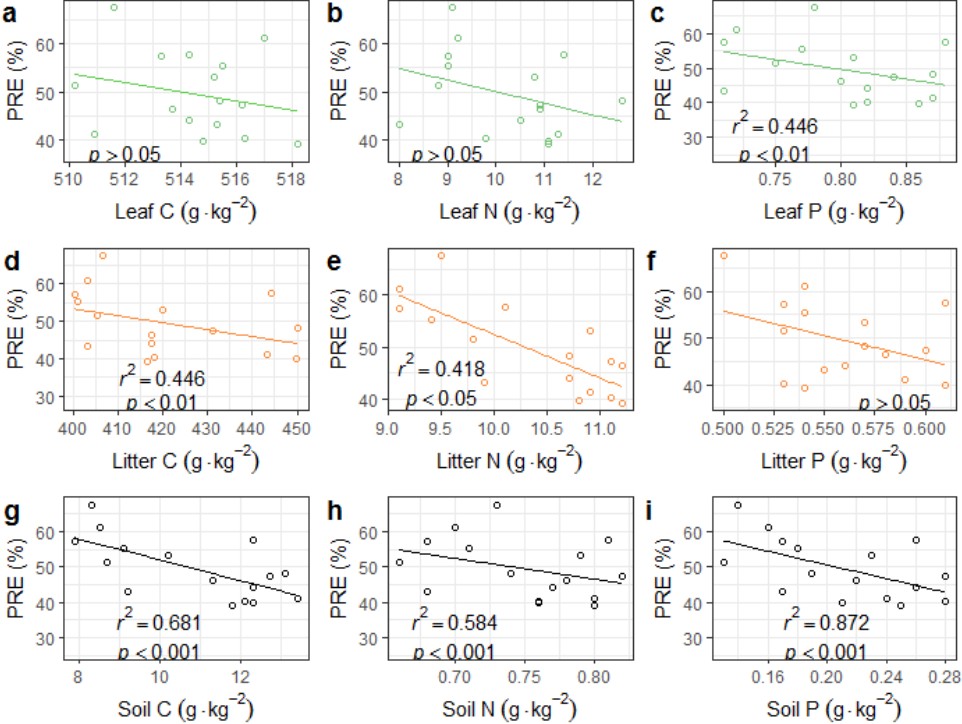

**Figure A3.** Relationships between PRE (P resorption efficiency) and leaf (**a**–**c**), litter (**d**–**f**), and soil (**g**–**i**) C, N, and P contents.

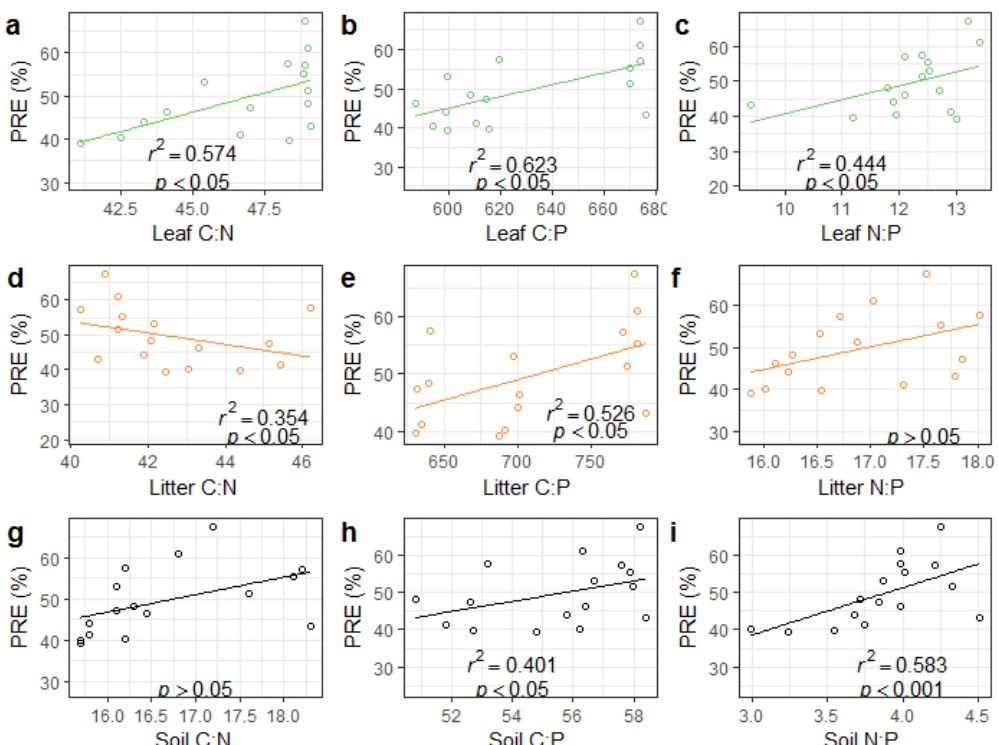

**Figure A4.** Relationships between PRE (P resorption efficiency) and leaf (**a–c**), litter (**d–f**), and soil (**g–i**) stoichiometry.

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
