# Peer review of "Nutrient Resorption and C:N:P Stoichiometry Responses of a Pinus massoniana Plantation to Various Thinning Intensities in Southern China"

_forests, doi:10.3390/f13101699_

Round 1
Reviewer 1 Report
Dear authors,
I congratulate you on your work and the elaborated manuscript.
I would only have two observations to make:
- a more detailed soil analysis was recommended (the content of humic and fulvic acids)
- the level of development of mycorrhizae (A.M), knowing that they have a major impact on nutrition.
Congratulations once again for the idea of studying this topic!
Author Response
Reviewer 1 comments
Point 1. a more detailed soil analysis was recommended (the content of humic and fulvic acids)
Response1: Thanks to the reviewer for further advice, indeed the content of humic and fulvic acids is an important factor, the reason for the amount of citation text task, and then for the forest management design of Guangxi Masson pine, In this study, we aimed to examine the mechanisms underlying the impacts of thinning on C:N:P stoichiometry in a Pinus massoniana plantation, focusing on interactions among soils, plant tissues (leaves and litter), and soil properties. We conducted four different thinning treatments to determine the effects of thinning on the C:N:P stoichiometric ratios in leaves, litter, and soil in a Pinus massoniana plantation ecosystem. In addition, these differential responses suggest that soil physicochemical processes play a crucial role in regulating the effects of thinning. The N:P ratios of green leaves and litter and the nutrient resorption trends demonstrated that N was the limiting element for Pinus massoniana growth on plantations in the study region. Thinning intensity regulates the biogeochemical cycles of C, N and P in Pinus massoniana plantation ecosystems by affecting nutrient resorption and soil physicochemical processes. The inconsistent results obtained can be attributed to the complexities of stand environments and the redistribution of site resources following thinning. Therefore, incorporating the effects of thinning intensity into nutrient cycling models may improve predictions related to achieving long-term forest management strategies.
Point 2: the level of development of mycorrhizae (A.M), knowing that they have a major impact on nutrition.
Response2: Thanks to the reviewer for further suggestions, ecological stoichiometry is the study of the proportional relationship of chemical elements in the ecological process and its change with biological and non-biological environmental factors. The study of the spatio-temporal distribution patterns of carbon (C), nitrogen (N) and phosphorus (P) stoichiometry has become a hot topic. The study of soil C:N:P characteristics lagged behind that of plants, and the universal spatio-temporal distribution law had not yet been obtained, and the environment-driven mechanism of spatio-temporal variation was less involved. At the same time, as the supply environment of forest tree growth nutrient resources, the C:N:P of soil can not only indicate the decomposition and nutrient status of soil organic matter, but also be closely related to the biomass allocation, diversity and ecological function of forest communities. Therefore, the spatio-temporal variation pattern and driving mechanism of forest soil C:N:P can effectively reveal the response of forest ecosystems to environmental gradients and the changes in their ecological functions in the spatio-temporal dimension. For the mycorrhizae (A.M) study, we will be conducting further logging experiments, thank you again to the reviewers for their constructive comments.
Reviewer 2 Report
It is a ground-breaking work with a great deal of research. It provides some very relevant data for forest management. However, more research is needed on soil life, especially on mycorrhizal fungi, in order to understand the relationship between plants and nutrient cycles.

Author Response
Dear editor and reviewer:
I am very grateful to your comments for the manuscript. According with your advice, we amended the relevant part in manuscript. According to the reviewers' comments, we have revised the manuscript to provided our explanation. Any revisions made to the manuscript should be marked up using the“Track Changes” function. We have checked that all references are relevant to the contents of the manuscript. Before submitted, we have use the mdpi editing services listed at https://www.mdpi.com/authors/english in English writing and . Furthermore, we revised the paper according to your suggestion. Some of your questions were answered below.
I hope you are satisfied with the revised version.
Thank you.
Reviewer 2 comments
Point 1: It is a ground-breaking work with a great deal of research. It provides some very relevant data for forest management. However, more research is needed on soil life, especially on mycorrhizal fungi, in order to understand the relationship between plants and nutrient cycles.
Response 1:: Thanks to the reviewer for further advice! indeed, the content of humic and fulvic acids is an important factor, the reason for the amount of citation text task, and then for the forest management design of Guangxi Masson pine, In this study, we aimed to examine the mechanisms underlying the impacts of thinning on C:N:P stoichiometry in a Pinus massoniana plantation, focusing on interactions among soils, plant tissues (leaves and litter), and soil properties. We conducted four different thinning treatments to determine the effects of thinning on the C:N:P stoichiometric ratios in leaves, litter, and soil in a Pinus massoniana plantation ecosystem. In addition, these differential responses suggest that soil physicochemical processes play a crucial role in regulating the effects of thinning. The N:P ratios of green leaves and litter and the nutrient resorption trends demonstrated that N was the limiting element for Pinus massoniana growth on plantations in the study region. Thinning intensity regulates the biogeochemical cycles of C, N and P in Pinus massoniana plantation ecosystems by affecting nutrient resorption and soil physicochemical processes. The inconsistent results obtained can be attributed to the complexities of stand environments and the redistribution of site resources following thinning. Therefore, incorporating the effects of thinning intensity into nutrient cycling models may improve predictions related to achieving long-term forest management strategies.
Point 2: Line 79: scientific name
Response 2:: corrected. See line 80.
Point 3: Line 92 and other factors not taken into account so far, such as soil micro-organisms or mycorrhizal fungi.
Response 3:: corrected. See line 97.
Point 4: Line 132. use an international classification such as the WRB
Response 4: revised. See line 133-134. The soils at the field site are typical Cambisols . Added the references: IUSS Working Group WRB, 2006. World reference base for soil resources. World Soil Resources Reports No. 103, 2nd edition. Food and Agriculture Organization (FAO). Rome.
Point 5: Line 134-137 this information is not necessary, delete
Response 5:: deleted. See line136-140.
Point 6: Line 157: what is the difference between plot and quadrat?
Response2: It is no differences. We has been unified. See line 162.
Point 7: Line 164: Explain better, how much soil was collected, when (after thinning, before, ...) and how?What does 2.5cm in diam?
Response 7: Thank you to the reviewers for their valuable comments. We have revised these suggestions. See line 166-169.
Point 8: Line 208 :An explanation of when the soil was collected is necessary as the values of soil elements can vary greatly from point to point even in the same soil type.
Response 8: Thank you to the reviewers for their valuable comments. “when the soil was collected” Already added. In September 2019 (the period of vigorous forest growth). See line172.
Point 9: Line237: A direct relationship is seen between the values of these elements in the soil and in the leaves of the trees. However, it does not necessarily have to be related to thinning. The soil values before and after thinning should be known.
Response 9: The protective function of forests has weakened and has a significant impact on the destruction of forest ecosystems. In the process of logging, large-scale mechanized operation methods, timber transportation, etc. will cause different degrees of damage to forest vegetation and soil. This is because the harvesting intensity and the way of transporting wood will destroy the soil, so that the exchange between nutrients, moisture and air in the soil is blocked, the activity of soil microorganisms and fungi is limited, and the soil fertility is seriously degraded. Therefore, reasonable measures should be taken in this case, so that scientific forest harvesting can promote tree growth, improve forest stands, facilitate forest regeneration and restoration, maintain animal and plant diversity and sustainable forest management. And we added explanations on the relation to thinning, and some references to empirically prove. See line 394-395.We believe that our study provide important insights for improving forest management practices to maintain ecosystem functioning in forest ecosystems and should be interesting to international readers of the prestigious journal.
